# Position: Lottery Tickets Are Misleading! Use Escape Dimensions to Explain The Success of Overparameterization

## Abstract

The lottery ticket hypothesis is often used as a didactical analogy to explain the success of overparameterized neural networks: "larger networks succeed because they more likely contain a well-initialized subnetwork that can learn the task in isolation, much like buying more tickets increases the chances of winning a lottery." This explanation is intuitive but misleading: it suggests that subnetworks can be treated independently from the rest of the network. Following this reasoning leads to interpreting learning in wide networks as a multi-start optimization process, where gradient descent simply conducts a parallel search over subnetworks. We argue that this view is flawed since, among other reasons, winning tickets can be made to fail by perturbing the rest of the network. We put forward a more accurate intuitive picture for the success of overparameterized networks based on the geometry of the loss landscape: increasing width expands the set of available dimensions for optimization, making it easier to escape bad local minima. Moreover, as width grows, bad minima become increasingly rare relative to good minima, leading to a higher likelihood of convergence to good solutions. As the field grows more mature, it is important to refine the analogies we use to explain foundational phenomena, such as the apparent redundancy of large networks, reconciliating practitioners' intuitions with modern theoretical insights.

## 1. Introduction

Nowadays, we can optimize neural networks to achieve impressive results by implementing gradient descent with a few lines of code. Yet, behind this simplicity lies the coordi-

[1]Anonymous Institution, Anonymous City, Anonymous Region, Anonymous Country. Correspondence to: Anonymous Author <anon.email@domain.com>.

Preliminary work. Under review by the International Conference on Machine Learning (ICML). Do not distribute.

---

**Escape Dimensions Theory**

Adding seemingly redundant parameters to a neural network increases the number of dimensions available to escape sub-optimal minima, making it easier for gradient descent to traverse the loss landscape.

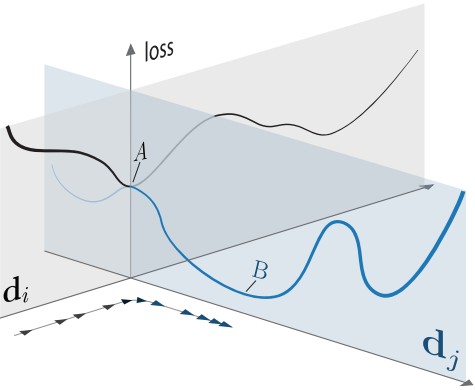

*Figure 1.* In the landscape spanned by the axis $\mathbf{d}_i$, gradient descent approaches the local minimum $A$ (black path). With overparameterization, the new landscape dimension $\mathbf{d}_j$ (in blue) enables escape to a better minimum $B$. Actual path to minimum may later drift away from the axis $\mathbf{d}_j$.

nation of billions of moving parts: the network parameters. These parameters interact in complex ways to arrange themselves into configurations that solve challenging tasks. The lottery ticket hypothesis (LTH) (Frankle & Carbin, 2018) has become a popular way to make sense of this complexity, by stripping down the many nonlinear interactions to a simple combinatorial picture of competing subnetworks: each subnetwork is a lottery ticket, and training a large network is like buying many tickets to increase the chances of finding the one that can solve the task. The field has embraced this metaphor to explain why seemingly redundant, large networks are necessary for training, permeating both research and didactical content. But is this analogy accurate? In this piece, we argue that **the lottery ticket analogy is a misleading mental model for understanding overparametrization in neural networks** and that instead **the geometry of the loss landscape provides a more accurate intuition, via the concept of Escape Dimensions** (Fig. 1).

## 2. A common wisdom in the field

*While directly identifying lottery tickets before training remains impractical, the underlying ideas have significantly influenced our explanatory theories of sparsity, pruning, and network efficiency.* [Q23]

Jeffares & van der Schaar (2025)

Before the Lottery Ticket Hypothesis (Frankle & Carbin, 2018) was proposed, countless experiments in the pruning literature had shown that large models, once trained, could be stripped of most of their parameters without significant loss in performance (LeCun et al., 1989; Reed, 1993; Han et al., 2015; Srinivas & Babu, 2015; Hoefler et al., 2021). In 2018, Frankle and Carbin discovered that there exist sparse networks (subgraphs) that can be trained from scratch to full accuracy, given the right initialization. The existence of these so-called *winning tickets* was surprising, because it seemed that dense, large networks were necessary for successful training. However, identifying these well-initialized subgraphs still required information from training a dense network. Since then, the LTH inspired a large body of work, with the dream of reducing the energy demands of deep learning not only at inference, but also at training time (Liu et al., 2024). The observation that small networks are difficult to train, despite the *existence* of winning tickets, together with the LTH popularity, created fertile ground for the lottery analogy to spread as an intuitive explanation of why larger networks are easier to optimize. None of the arguments we present in this paper challenge the empirical validity of LTH. What we challenge is the popular intuition, or analogy, or common wisdom that LTH has generated in the community to explain the apparent redundancy of large neural networks. The analogy of tickets and lotteries implies three specific properties:

- **Independence**: tickets in a lottery are independent samples.

- **Sufficiency**: having a winning ticket is sufficient to win the lottery.

- **Scaling**: the probability of winning the lottery scales predictably with the number of purchased tickets.

We find that, over the literature and in various forms of didactical content, these properties are often directly or indirectly translated to the context of subnetworks embedded in overparameterized network models (see section 2.2 and Tables 1, 2). Namely:

- **Independence**: the training outcome of a subnetwork is independent from the rest of the network.

- **Sufficiency**: containing a winning subnetwork at initialization is sufficient to train a network successfully.

- **Scaling**: the probability of successful training scales combinatorially with the size of the network. [Q30]

If these ideas were correct, training large networks would amount to running many subnetworks in parallel, with the successful one emerging as the winner over the course of optimization. In the rest of this section, we will clarify the origin of the misconceptions, provide evidence for how widespread they are in the literature and in online content, and describe how learning in overparameterized networks would work if the lottery analogy would be interpreted literally.

### 2.1. Mistaking the hypothesis for the conjecture

*"Empirically, many people have found that bigger models are easier to train (often explained with the 'lottery ticket hypothesis')"* [Q5]

Abnar et al. (2020)

The statement that defines the original hypothesis is of empirical nature: it hypothesizes the existence of winning subnetworks inside large, dense, successfully trained networks. This phenomenon has been thoroughly validated (Frankle & Carbin, 2018; Zhou et al., 2019; Morcos et al., 2019; Renda et al., 2020; Ma et al., 2021) and extended, with some variants, in a vast range of settings (Frankle et al., 2020). What holds empirically is described in the original statement of the hypothesis:

***The Lottery Ticket Hypothesis.*** *A randomly-initialized, dense neural network contains a subnetwork that is initialized such that — when trained in isolation — it can match the test accuracy of the original network after training for at most the same number of iterations.*

The intuition behind the analogy, however, is also tightly linked to the conjecture that they put forward in order to provide a causal explanation for their observations:

***The Lottery Ticket Conjecture.*** *[...] SGD seeks out and trains a subset of well-initialized weights. Dense, randomly-initialized networks are easier to train than the sparse networks that result from pruning because there are more possible subnetworks from which training might recover a winning ticket.*

The authors of the LTH were careful in claiming that their conjecture was not supported by any empirical evidence. We speculate, without blaming the original authors, that the choice of words (lottery, tickets, winning) may have

---

Superscripts of the form Q# refer to verbatim quotes from academic and online sources, collected in Appendix Tables 1 and 2. These illustrate how the lottery metaphor is commonly expressed and interpreted.

contributed to the community absorbing the conjecture as if it were a more or less established phenomenon. We argue that the use of the analogy in the many subsequent papers in the field contributed to mistaking the evidence supporting the existence of winning tickets for evidence supporting the causal explanation that large networks learn *because* there exist winning tickets embedded in them.

In a fair lottery (for example, one run by a charity with the help of benevolent people), the number of tickets is set beforehand, and once all the tickets are sold we are sure that the winning ticket has been drawn. There is no possibility to buy more than the available tickets and no statement can be made of how the prize would change in a bigger lottery with more tickets. Similarly, Frankle & Carbin (2018) used a single large network that was successfully trained and the design of the simulation study made no statement of what would happen in a larger network. However, the reception of the lottery ticket paper focused on the scaling of success with network size: a larger network contains more subnetworks and "therefore" more tickets and is "therefore" more successful. It is no longer a charity lottery, since smaller networks are not guaranteed to come with a winning ticket; stochastic games (roulette, bandits) could, however, still be a useful analogy, as long as the attempts are independent. As we illustrate below in several examples, the mental picture of subnetworks as "independent attempts" is itself problematic.

### 2.2. How the lottery analogy is integrated in the field

*"If you start with a very overparameterized network, probability theory gives the network much higher chances to include a better subnetwork than a very small one."* [Q9]

Koster et al. (2022)

Statements of this form are often used to explain why large networks are needed to fit data successfully. They are usually invoked to resolve the apparent paradox that successful sparse subnetworks *do exist* and can be found via pruning, but *only after training* a large dense model. Similar statements appear across a wide range of sources, including research papers, blog posts, and lecture slides. Some of these sources explicitly endorse one of the three problematic ideas stated above (independence, sufficiency, scaling), others mention them as part of a broader narrative, while some even report these as hypothesis backed by evidence (and not conjectures). By scanning the literature, we found a collection of more than 30 representative quotes that we list verbatim in Tables 1,2 (Appendix). We describe our methodology for collecting them in Section A (Appendix).

This framing also manifests indirectly in the kinds of research questions it motivates. One notable example is the

theoretical and empirical search for conditions under which well-performing subnetworks exist at initialization. The paper titled "Proving the lottery ticket hypothesis: pruning is all you need" (Malach et al., 2020) formalizes and proves a modified version of the LTH, sometimes referred to as the *strong lottery ticket hypothesis*. This variant conjectures that subnetworks with good performance can exist at initialization, without requiring any training at all (Ramanujan et al., 2020). Many subsequent works have built upon this idea, improving bounds and exploring various architectures and settings (Pensia et al., 2020; Orseau et al., 2020; Burkholz et al., 2021; Berner et al., 2022; Burkholz, 2022; da Cunha et al., 2022; Ferbach et al., 2022; Natale et al., 2024; Kumar & Natale, 2025; Otsuka et al., 2025). These works are valuable in their own right and do not aim to explain why overparameterized networks are easier to optimize. However, they implicitly reinforce one particular framing of tickets and lotteries: success is associated with the existence of suitable subnetworks, rather than with the dynamics of learning in the full parameter space. Calling this variant a strong version of the LTH further encourages this interpretation, despite the fact that it no longer addresses training at all. In this view, success is explained entirely in terms of pre-existing structure, encouraging a combinatorial picture in which the task is to identify the right subnetwork. Even when such subnetworks provably exist, they are not available a priori as independent objects. Absurdly, but not surprisingly, identifying them in practice requires another dense, overparameterized optimization process. The same that the original motivation hoped to avoid (Zhou et al., 2019; Ramanujan et al., 2020; Wortsman et al., 2020; Bai et al., 2022b).

Finally, this interpretation is also familiar from informal scientific discourse. In our own experience, we hear and see it widely used in talks, discussions, and pedagogical settings to provide an intuitive account of why overparameterization is helpful. At the time of this piece, many readers will likely recognize similar explanations from their own interactions.

### 2.3. Multi-start optimization?

*"If you want to win the lottery, just buy a lot of tickets and some will likely win. Buying a lot of tickets = having an overparameterized neural network for your task."* [Q27]

Princeton-CS-598D (2020)

In a lottery, it is enough to have one winning ticket to win the prize; all other tickets are irrelevant. In discussions related to the LTH, one frequently encounters expressions such as "*larger networks are more likely to contain a winning ticket*"[Q1Q2Q4Q19], or "*SGD seeks out a winning ticket*"[Q10Q24Q26] or "*width acts as a form of parallel search*"[Q18Q28Q31]. When stated by various authors, these formulations rarely fully commit to the claims.

However, they are not conceptually neutral. Taken seriously, they suggest that subnetworks are well-defined objects at initialization, whose optimization success is fixed prior to training, and that optimization acts primarily as a process of selection. Once this interpretation is adopted, the implied picture of learning becomes quite specific: if subnetworks are meaningful candidates that pre-exist training, and if success requires only one of them to be "good," then optimization (SGD) can be understood as a process of selection among these candidates. Training no longer plays a role in optimizing a solution; it merely enables the optimization trajectory of the winning subnetwork while suppressing the others[Q18]. From this perspective, overparameterization helps by increasing the number of independent attempts at solving the task. Fig. 2 illustrates this view. Because the number of subnetworks available in a dense network grows combinatorially with network width, this view *predicts* an extremely rapid increase in the likelihood of success as models become larger. Despite its intuitive appeal, this implication is incorrect, as we will show in the next section.

## 3. Tearing winning tickets apart

*"the chance of any given ticket winning is tiny, but if you buy enough of them you are certain to win, and the number of possible subnetworks increases exponentially as the power set of the set of connections"* [Q30]

Wikipedia (2024)

Let us now examine the implications of the sufficiency, scaling and independence properties more closely, and make testable predictions. Given an initialization vector $I \in \mathbb{R}^d$, we denote by $R_I \in \{\texttt{fail}, \texttt{success}\}$ the outcome of training the entire network. Similarly, $R_I^{(n)}$ is the outcome of training subnetwork $n$ in isolation: this is obtained by taking the dense network initialized with the same $I$, and training it while applying a pruning mask $m_n \in \mathbb{R}^d$ to its graph and initialization, $I \odot m_n$. The sufficiency property states that if there exists at least one winning ticket in the initialization, then the dense network will succeed:

$$\exists n \mid \{R_I^{(n)} = \texttt{success}\} \implies \{R_I = \texttt{success}\} \quad (1)$$

Does this **sufficiency** property hold in general? We can design a simple experiment to test it. Let us take a dataset generated by a two-layer MLP with 4 hidden neurons: $y(\mathbf{x}) = \sum_1^4 a_i \text{ReLU}(\mathbf{w}_i^\top \mathbf{x} + b_i)$ and use it to train a two-layer MLP with $r = 4$ hidden neurons by minimizing the mean squared error loss over a set of 30k samples. Less than 15% of the random initializations led to successfully learning the generator network (final loss $< 10^{-20}$), all other solutions hovered around a final loss of $10^{-2}$ (details in Appendix B). In the spirit of the LTH philosophy,

Original parameter space      Space of each subnetwork

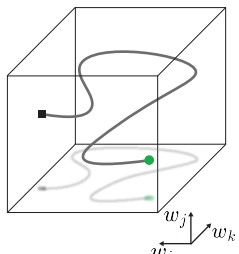 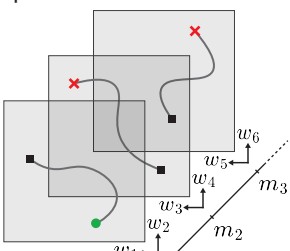

*Figure 2.* **Multi-start optimization view of the landscape.** Since the outcome of a lottery ticket is independent from other tickets, the analogy *implies* that each subnetwork's optimization outcome is independent. Seen from the parameter space perspective, it is as if we could describe the trajectory of learning in a large space (left) as a combination of independent trajectories over subspaces (right). Each slice of parameter space is defined by a subnetwork weight mask $m_i$. If true, training an overparameterized network would be equivalent to running multiple independent optimizations in parallel, each corresponding to a differently initialized subnetwork.

we select the successful initializations and define them as our winning ticket initializations $I^*$. We then set up a new two-layer MLP with $r > 4$ hidden neurons, and initialize it such that the first 4 neurons equal the winning ticket initialization $I^*$ (green subnetwork of Fig. 3), while the additional neurons are initialized from a Glorot normal distribution $\mathcal{G}$ (Glorot & Bengio, 2010), namely $I = [I^*, I^+ \sim \mathcal{G}]$ (black neuron(s) in Fig. 3). We now have a setup where $\exists n \mid \{R_I^{(n)} = \texttt{success}\}$. Based on eq.1, the larger network should also succeed, i.e. $\{R_I = \texttt{success}\}$. However, embedding this winning ticket in a network with one additional neuron ($r = 5$) yields success in only $\sim 65\%$ of runs, sharply contradicting the sufficiency prediction (100%) of eq. 1. Fig. 3-top shows how the success rate of larger networks evolves as neurons are added to a winning ticket (dark bars), compared to training from random initialization (light bars). In contrast, the sufficiency property of the lottery intuition in eq.1 would predict a 100% success rate (empty dashed green bars). These results show that initializing a larger network with a winning ticket embedded in it does not guarantee successful training.

From eq.1 we can derive a prediction for the **scaling** of the probability of success with the number of subnetworks embedded in a dense network. We will use the interpretation of lottery related to the conjecture (Section 2.1): subnetworks are subgraphs of the network together with a given initialization. Because of the random initialization, the pool of potential subnetworks is infinite while the subgraphs increase combinatorially with network size. By the properties of logical implication, the contrapositive of eq.1 also holds:

$$\{R_I = \texttt{fail}\} \implies \{R_I^{(n)} = \texttt{fail}\} \, \forall n \quad (2)$$

In general, the training result $R$ depends on many factors, some stochastic (e.g., data shuffling) and some deterministic

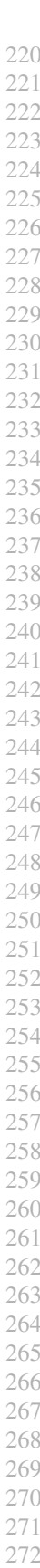

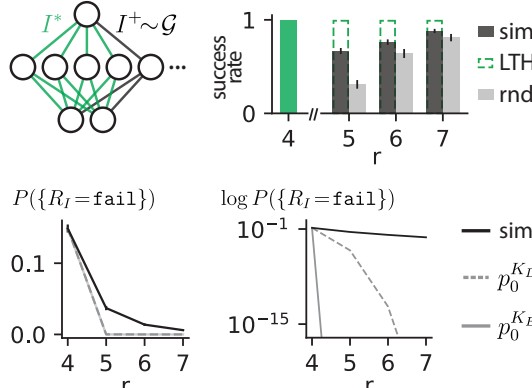

*Figure 3.* **Subnetworks do not respect the properties of a lottery.** *Top*: The histogram gives the success rate (dark bars) for networks of size $r > 5$ in simulations where the depicted network is initialized with the winning ticket $I^*$ (green connections) and overparameterized with randomly initialized neurons $I^+$ (black). We compare to randomly re-initializing the entire network (light bars). The presence of the winning ticket inside the network would predict a success rate of 1 (LTH, open green bars). *Bottom*: the probability of failure of a large network does not decrease combinatorially with the number of subnetworks embedded in it. Note the y-axis: the scaling predicted by the combinatorial intuition given by the LTH is totally incorrect.

(e.g., architecture, optimizer, dataset). When accounting for all these and drawing an initialization from a distribution, e.g. $I \sim \mathcal{G}$, we can denote the probability of success of the dense network as $P(\{R_I = \texttt{success}\})$. Making use of eq. 2 and the independence property, we can write:

$$P(\{R_I = \text{fail}\}) = P\left(\bigcap_{n=1}^{N} \{R_I^{(n)} = \text{fail}\}\right) \quad (3)$$

$$= \prod_{n=1}^{N} P\left(\{R_I^{(n)} = \text{fail}\}\right) \quad (4)$$

which predicts that the probability of failure of a large network is the product of the probabilities of failure of all its subnetworks. Or more loosely, that the probability of failure decreases exponentially with the number of subnetworks embedded in the dense network. With a similar teacher-student setup, where success is unequivocally defined by zero training loss, we can measure the scaling of $P(\{R_I = \texttt{fail}\})$ with the number of subnetworks $N$ and compare it to the scaling of eq.4. We first generate a dataset from a two-layer MLP with 4 hidden neurons, then train two-layer MLPs with varying number of hidden neurons $r$, from 4 to 7. Hence, the smallest network that can learn the task, i.e. the winning ticket, contains 4 hidden neurons. The number $N_D(r)$ of size-4 dense subnetworks that can be embedded in a size-$r$ network is given by standard combinatorics: $N_D(r) = \binom{r}{4}$. In a second set of simulations, we measure the probability $p_0$ of failure of a typical "size 4" network in *isolation* by training 4000 MLPs with $r = 4$ hidden neurons. Note

that, by definition, $p_0 = P(\{R_I^{(n)} = \texttt{fail}\})$ since each "size 4" network is a potential subnetwork in a bigger dense network. Then, according to the prediction of equation 4, the probability of failure of larger networks ($r > 4$) should scale as $P(\{R_I = \texttt{fail}\}) \propto p_0^{N_D}$. If we consider subnetworks defined in terms of edges, and not neurons, the decay is even more pronounced: let $N_E(r)$ be the number of all subnetworks with as many edges as the teacher network, then $N_E(r) = \binom{r(d_{\text{in}}+d_{\text{out}})}{4(d_{\text{in}}+d_{\text{out}})}$. This would produce an even faster scaling $\propto p_0^{N_E} \approx p_0^{(d_{\text{in}}+d_{\text{out}})N_D}$. Lastly, we estimate $P(\{R_I = \texttt{fail}\})$ by training 4000 networks for each width $r > 4$. Fig. 3-bottom shows the predicted decay of $P(\{R_I = \texttt{fail}\})$ (grey lines) in linear and log scale. The prediction disagrees with the simulation results (black trace). Hence, any reference to the combinatorial power of subnetworks to explain the success of width is misleading.

**Does it make sense to talk about subnetworks independently?** There are two fundamental reasons for doubting this: (i) Subgraphs are composed of a set of neurons with a specific connectivity mask. When considering subnetworks of non-trivial size, different subgraphs necessarily share neurons. This limits the independence of their initializations. In other words, initializations of neurons are independent, but initializations of subgraphs are not. (ii) Even if we have identified a winning subnetwork, its gradient depends on the entire network. Take a subnetwork $\hat{y}(\mathbf{x})$ (of arbitrary size) and consider a generic loss $\mathcal{L} = \frac{1}{N} \sum_n^N c(y_n, \hat{y}(\mathbf{x}_n))$, where $c(\cdot, \cdot)$ is a per-sample cost function (e.g., cross-entropy, MSE) and $y_n$ is the label of the $n$-th sample. We can compute the gradient of the loss with respect to the activations of a hidden layer $\mathbf{h}_l$ as:

$$\frac{\partial \mathcal{L}}{\partial \mathbf{h}_l} = \frac{1}{N} \sum_{n=1}^{N} \left(\frac{\partial \hat{y}(\mathbf{x}_n)}{\partial \mathbf{h}_l(\mathbf{x}_n)}\right)^\top \nabla_{\hat{y}} c(y_n, \hat{y}(\mathbf{x}_n)) \quad (5)$$

Let us add an extra neuron to an arbitrary hidden layer $m \neq l$. The subnetwork $\hat{y}$ is now embedded in a slightly larger network $\hat{y}^+$. The gradient of the hidden layer $l$ belonging to the original subnetwork $\frac{\partial \mathcal{L}}{\partial \mathbf{h}_l}$, is now dependent on the *entirety* of the larger network for two reasons: first, for most choices of cost functions $c(\cdot, \cdot)$, the gradient factor $\nabla_{\hat{y}^+} c(y_n, \hat{y}^+(\mathbf{x}_n)) \in \mathbb{R}^{d_{\text{out}}}$ is a function of $\hat{y}^+$; second, because of backpropagation, the higher dimensional term $\frac{\partial \hat{y}^+(\mathbf{x}_n)}{\partial \mathbf{h}_l(\mathbf{x}_n)} \in \mathbb{R}^{d_{\text{out}} \times d_l}$ will depend on the new neuron's activation due to the forward path for $m < l$ and $m > l$, and on the new neuron's parameters for the backward path for $m > l$. A similar argument holds when considering adding individual edges.

Despite the widespread use of the lottery analogy, our analysis shows that each of the explicit or implicit assumptions of lottery tickets are not only theoretically questionable, but also invalidated by empirical evidence.

*Figure 4.* **Sketched illustrations of loss landscapes as overparameterization increases.** Overparameterization increases from left to right. While the illustrations of in this figure are useful to convey a sense of the types and proportions of minima in the landscape, it is important to note that they do not accurately represent the geometry of the landscape. For example, they mis-represent the density of critical points with respect to the entire volume (they likely are a zero-measure set of the parameter space).

## 4. Escape Dimensions

The mental picture of lottery tickets is oversimplified because it neglects training dynamics, parameter interactions, and the geometry of the loss function. Rather than adopting a mental model detached from the space where learning actually happens, we should build our intuition on the loss landscape itself. Let us ask **what** the landscape looks like **and** *how* the width of a hidden layer changes it.

In response to the *what* question, there exists a large body of theoretical work characterizing features related to convergence and trainability. The two classic pictures of loss landscapes correspond to two opposite regimes of width. In the classic regime (i.e., networks with "narrow" hidden layers), landscapes are often pictured as rugged, with many poor local minima that can trap optimization (Fig. 4, first panel). The landscape is dominated by high loss local minima, good solutions exist, but they are difficult to find as their basins of attraction are small. The presence of bad local minima has been widely discussed theoretically (Auer et al., 1995; Safran & Shamir, 2018) and verified emprically in practical settings (Martinelli et al., 2024; 2025); in fact, finding the optimal weights in this regime is NP-hard (Blum & Rivest, 1988; Livni et al., 2014). In the vastly overparameterized regime (Safran & Shamir, 2016; Freeman & Bruna, 2016; Bahri et al., 2020; van Meegen & Sompolinsky, 2025), the hidden layers are wide and contain so many neurons that the number of parameters exceeds the number of training samples (*interpolation* regime (Belkin, 2021)). The loss landscape, despite still non-convex, is benign: training almost always converges to global minima (Belkin, 2021; Liu et al., 2022), in line with the neural tangent kernel theory (Jacot et al., 2018). The fourth panel gives an intuition: random initializations start at a high loss (spikes), but *training* leads to a global minimum, corresponding to a "sea" of zero-loss solutions (van Meegen & Sompolinsky, 2025). Taken together, these results show that the loss landscape looks very differently in narrow and wide regimes. In particular, the loss landscape of a wide network is not just a larger instance of that of a narrow network, but *qualitatively* different.

### 4.1. Increasing width transforms minima into saddles

How can increasing width induce a qualitative change in the loss landscape? In the following, we argue for an intuition, or mental model, to explain *why* overparameterization enables successful training: **escape dimensions** (Fig. 1). Escape dimensions provide a geometric perspective that is complementary to the convergence results aforementioned, by offering an intuitive picture of what changes when we say "*width reshapes the loss landscape*". Contrary to the LTH with subnetworks, we ground our intuition in theoretically tractable objects: **critical points**, i.e., points in parameter space where the gradient is zero. In high-dimensional spaces, it is easy to lose geometric intuition: visualizations obtained from random projections (Goodfellow et al., 2014; Im et al., 2016; Li et al., 2018) are difficult to relate to mathematical structures relevant to learning. We can think of critical points as landmarks for orienting ourselves in high dimensions. These geometrical anchors are meaningful because they shape learning trajectories, much like mountain passes shape hiking routes in mountainous terrain. We focus on critical points of the loss function, in particular, minima and saddle points with many positive curvature directions. The seminal work of Fukumizu & Amari (2000) shows that any critical point of a network with $r$ hidden neurons is also a critical point of any larger network with $m > r$ hidden neurons. In particular, if a critical point $\hat{\theta}$ is a local minimum in the smaller network, addition of one (or more) neurons transforms it into a set of critical points $\hat{\theta}^+$ in the larger one:

$$\nabla_\theta \mathcal{L}_r(\hat{\theta}) = 0 \implies \nabla_{\theta^+} \mathcal{L}_{m>r}(\hat{\theta}^+) = 0 \qquad (6)$$

where the transformation $\hat{\theta} \to \hat{\theta}^+$ is performed in a way to maintain functional equivalence between the two networks. We call this transformation *neuron splitting*. Intuitively, a neuron is added by copying the input weight vector (and bias) of one of the neurons in the network of size $r$, and sharing its output weights with relative re-scaling $\gamma$ and $(1 - \gamma)$ between the old and the new neuron. Neuron splitting is of fundamental importance because it means that adding neurons to a network may transform the nature of

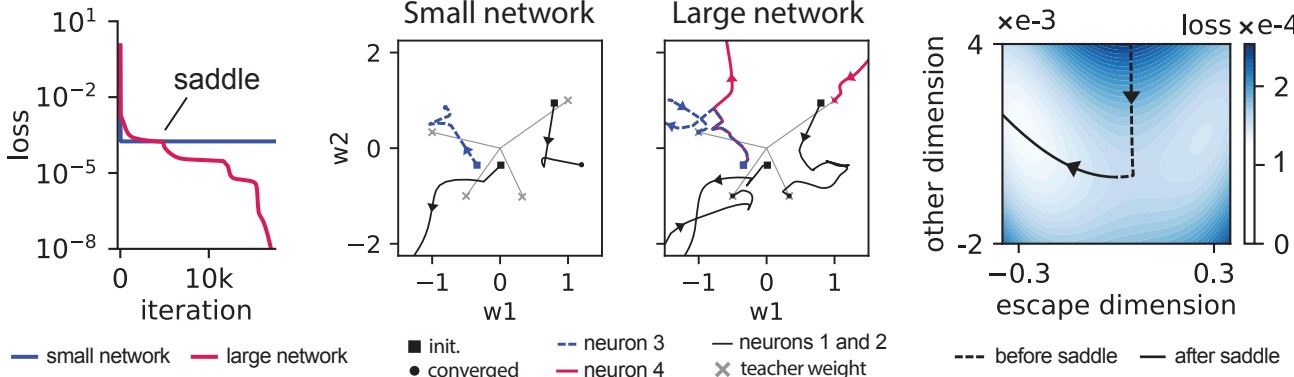

*Figure 5.* **Visualizing escape dimensions in a nonconvex landscape.** A 3-neuron and a 4-neuron two-layer MLPs are trained on data from a 4-neuron teacher, with shared initialization except for a split neuron, whose two copies form neurons 3 and 4 in the wider network. *Left*: loss curves: we expect to find a saddle point when the large network's loss crosses the small network's minimum. *Middle*: evolution of input weights: the two split neurons (blue and red traces) follow identical trajectories until the saddle point, after which they specialize and all neurons converge to the teacher weights (grey crosses). *Right*: loss landscape of the large MLP around the loss level of the small MLP's minimum. The escape dimension is defined by the direction of most negative curvature, the other direction is the one of highest curvature. The trajectory in parameter space is first orthogonal to the escape dimension (dashed line) until the critical point is surpassed via the escape dimension (continuous line).

its critical points from local minima to saddle points, by introducing **escape dimensions**. Formally, we define an escape dimension $\mathbf{e}_i^-$, a direction in parameter space along which the Hessian of the loss (computed at a critical point) has negative eigenvalues:

$$H\mathbf{e}_i^- = \lambda_i \mathbf{e}_i^-, \quad \lambda_i < 0 \tag{7}$$

where $H = \nabla^2_{\theta^+}\mathcal{L}(\theta^+)$ is the Hessian of the loss $\mathcal{L}$ with respect to the parameters $\theta^+$. Moreover, this fact defines a hierarchy of critical points across widths: a landscape of a network of width $m$ contains a subset of critical points (critical manifolds) that correspond to critical points of a smaller network with width $r < m$; hinting at a combinatorial, recursive structure of critical points across widths (Simsek et al., 2021; Zhang et al., 2025). The literature characterizes the stability of these critical manifolds: some sections are locally attractive, containing non-strict saddle points referred to as *plateau saddles* (Wei et al., 2008; Martinelli et al., 2025), other parts are repulsive, corresponding to *strict saddles* (Wei et al., 2008; Fukumizu et al., 2019; Wu et al., 2019; 2020; Simsek et al., 2021; Zhang et al., 2021; Petzka & Sminchisescu, 2021; Martinelli et al., 2025). In realistic settings, these manifolds almost always contain at least a few negative curvature directions, enabling escape from critical regions via gradient-based optimization (Petzka & Sminchisescu, 2021; Martinelli et al., 2025). Note that flat saddle regions can slow down or even trap learning dynamics under certain conditions (Inoue et al., 2003; Lee et al., 2016; Du et al., 2017; Chen et al., 2023).

To consolidate our intuitions, we consider the numerical example of a realistic, nonconvex landscape (Fig. 5). A smaller student network with three hidden neurons is trained to approximate a four-neuron teacher (details in B.3) and

converges to a local minimum with MSE of $\sim 10^{-4}$ (left). Because the input is two-dimensional, the evolution of each neuron's input weights can be visualized directly in the plane (middle). As expected, the three-neuron network cannot exactly reproduce the teacher (gray crosses). Starting from the *same initialization*, we apply a neuron-splitting operation to the original third neuron. The split replaces it with two duplicate neurons (neurons 3 and 4 in the "large network") that share identical input weights and have *nearly equal* output weights ($\gamma = \frac{1}{2} + \epsilon$), such that their combined contribution exactly matches that of the original neuron. Because neurons 3 and 4 are nearly identical and together functionally equivalent to the original neuron, the initial training dynamics of the wider network closely mirror those of the smaller one[1]. In the middle right panel of Fig. 5 neurons 1 and 2 (black) follow trajectories nearly identical to those in the smaller network, while neurons 3 and 4 (blue and red) track the trajectory of the original third neuron. At some point, the perturbation between the two neurons is amplified, causing their trajectories to diverge (top left sector). This occurs as the wider network passes through the neighborhood of the smaller network's local minimum (red and blue traces crossing in the left panel). The two neurons then diverge and specialize into distinct roles, allowing the wider network to escape the saddle inherited from the smaller model. This escape is made explicit by examining the local loss landscape of the wider network near the parameter values at which it matches the loss of the converged smaller network (saddle point in the left panel). The right panel shows a slice of the loss surface along the directions of minimum and maximum curvature of the loss. While approaching

---

[1]dynamics cannot be exactly identical, since the magnitude of the output weights affects the learning speed of the input weights.

the smaller network's lowest loss level (dashed line), the training trajectory descends primarily along the direction of maximum curvature (other dimension). After crossing this level, it bends and descends along the minimum-curvature direction, an escape direction made possible by neuron addition. Thus, adding neurons can create escape dimensions that transform local minima into saddles, enabling gradient descent to escape poor solutions to find better ones.

### 4.2. From local transformations to global structure

How do escape dimensions affect the global geometry of the loss landscape? Simsek et al. (2021) provide a global combinatorial perspective on increasing width by counting the saddle manifolds created through neuron splitting. These are named *symmetry-induced* saddle mandifolds, due to the inherent symmetry of the splitting process. When a task is realizable by a network of width $r$, any network of width $m > r$ contains a large number of manifolds of global minima at zero loss. These manifolds form a single interconnected web of valleys of global minima in parameter space. Both saddle manifolds and global minima manifolds proliferate combinatorially with increasing width, so growth rates alone are not informative. What changes is the balance between these structures: in the mildly overparameterized regime, the geometry is dominated by symmetry-induced saddle manifolds inherited from narrower networks (Fig. 4, second panel). As escape dimensions accumulate, the loss values of newly formed local minima decrease. However, as the network width increases further, a shift occurs: for $m \gtrsim 1.5r$, the number of global minima manifolds exceeds that of symmetry-induced saddle manifolds (Simsek et al., 2021). In other words, the landscape, defined by its critical points, is *dominated* by global minima. In this regime, it becomes increasingly likely for optimization to reach a global minimum (Fig. 4, third panel), consistent with empirical observations showing that bad minima become less prevalent as width increases (Martinelli et al., 2024). It is important to note that, depending on the dataset, the transition to a landscape dominated by global minima may occur before or after exceeding the number of parameters relative to the number of training samples. Escape dimensions emphasize the mental picture that, in a wide network, "all paths lead to the global minimum".

### 4.3. Escape Dimensions Theory and lottery tickets

The focus on the loss landscape also clarifies the role of lottery tickets. A winning ticket can be understood as an initialization of a *narrow network* that lies close to one of rare basins with small attraction radius in the rugged landscape (Larsen et al., 2021). A winning ticket is found by optimizing a *wide network* exploiting the benign structure of its landscape (panel 3). Escape Dimensions Theory explains why optimization becomes reliable as width increases. The existence of a winning ticket with good initialization *does not imply* that a (slightly) larger network containing the subnetwork (i.e., the subgraph with this specific initialization) leads back to the minimum. Because subnetworks are not independent. So why do the LTH simulations work? By construction, optimization starts from a large network, *successful* in finding a good minimum. Conceptually, the large network is now split into two parts: the "winning subgraph" and its complement. The possibility of a pruning step implies that the complement started with an initialization that enables its parameters to converge to have negligible effects (low magnitude). Thus, the winning network and its complement do not have an independent initialization. Results from a teacher-student setup generalize this point (Martinelli et al., 2024).

## 5. Alternative views and call to action

An alternative direction towards reasoning via subgraphs in neural networks is the notion of *neural race reduction* (Saxe et al., 2022; Li & Sompolinsky, 2022), that considers subnetworks as deep linear networks activated by individual input samples. Even if not strictly advocating for the LTH analogy, it shares the idea of isolating subnetworks to understand the whole network. Moreover, here we only addressed the question of trainability, leaving aside questions of generalization, benign overfitting, and implicit bias (Wilson, 2025). While escape dimensions are a strong candidate to explain the benefits of overparameterization, they are not the only actor at play, and their full implications remain to be explored.

As in all sciences of complex systems, understanding requires a reductionist approach. The subnetworks in the LTH are one example of such reduction, and this likely contributed to its popularity. Here, we have argued that using lottery tickets as an analogy in scientific discourse to explain overparameterization is misleading, because it is impossible to disentangle them from the rest of the system. Our representative but non-exhaustive selection of quotes (Tables 1, 2) indicates that this view is widespread in the community. We urge that the suggestive mental picture that the LTH evokes should not be reinforced in the literature nor taught in classes. Since mental images are important, we propose an alternative based on escape dimensions, grounded by theoretical evidence, but less adopted by the community. We suggest that the field should focus on understanding loss landscapes and training dynamics in a regime midway between classic narrow networks and vastly overparameterized networks. In our view, critical points are promising conceptual units to intuitively understand network computations. The study of saddle-to-saddle dynamics, and its interpretable outcomes are one such example (Jacot et al., 2021; Pesme & Flammarion, 2023; Kunin et al., 2025; Zhang et al., 2025).

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

*Table 1.* Quotes that suggest or reinforce the use of the lottery ticket hypothesis to explain the success of overparameterization

| | Quote | Paper |
|---|---|---|
| 1 | "and hypothesized that the role of overparameterization is to provide a large number of candidate subnetworks, thereby increasing the likelihood that one of these subnetworks will have the necessary structure and initialization needed for effective learning." | Mostafa & Wang (2019) |
| 2 | "the lottery ticket hypothesis, which argues that the probability of sampling a lucky, trainable sub-network initialization grows with network size due to the combinatorial explosion of available sub-network initializations" | Morcos et al. (2019) |
| 3 | "Hence, they showed that the pre-trained weights are not necessary, only the pruned architecture and the corresponding initial weight values." | Wang et al. (2020) |
| 4 | "they suggest that wide networks may perform as well as or better than thin ones because they 'buy more lottery tickets' and more reliably contain these fortuitously initialized subnetworks." | Casper et al. (2021) |
| 5 | "Empirically, many people have found that bigger models are easier to train (often explained with the 'lottery ticket hypothesis')" | Abnar et al. (2020) |
| 6 | "Surprisingly, VL+ performs better than VL , which might be because larger networks can help optimization of sub-networks VL $\subseteq$ VL+ as suggested by the Lottery Ticket Hypothesis" | Dubois et al. (2020) |
| 7 | "This polynomial bound already tells us that unpruned networks contain many 'winning tickets' even without training. Then it is natural to ask: could the most important task of gradient descent be pruning?" | Orseau et al. (2020) |
| 8 | "The hypothesis is that overparameterized neural networks are more likely to contain sub-network, which are initialized in such a way that they can be effectively optimized to solve the task." | Palm et al. (2021) |
| 9 | "If you start with a very overparameterized network, probability theory gives the network much higher chances to include a better subnetwork than a very small one." | Koster et al. (2022) |
| 10 | "This hypothesis suggests that "SGD seeks out and trains a well-initialized subnetwork" and that "overparameterized networks are easier to train because they have more combinations of subnetworks that are potential winning tickets." | Tripp et al. (2022) |
| 11 | "To explain why this happens, the lottery ticket hypothesis has been proposed [6], which posits that a randomly initialized model contains subnetworks that are especially suited for training on the given task (i.e. winning tickets), and that large models exponentially increase the chance of getting winning tickets." | Lam et al. (2022) |
| 12 | "In other words, as long as the dense network is initialized, the good subnetwork has been implicitly decided but needs to be revealed by the pruning process." | Bai et al. (2022b) |
| 13 | "the network redundancy also ensures a large random network contains a huge number of possible subnetworks, thus, carefully selecting a specific subnetwork should obtain promising performances. This point of view has been proved by [31, 44]." | Bai et al. (2022a) |
| 14 | "Similar to LTH, there is compelling evidence [38, 39, 14, 13, 2, 1, 45] suggesting that overparameterization is not essential for high test accuracy, but is helpful to find a good initialization for the network [30, 65]." | Verma et al. (2023) |
| 15 | "They theorize that in an overparameterized network, it is this subnetwork that effectively ends up being trained, thus preventing over-fitting. They also present a simple algorithm to identify this subnetwork." | Singh & Bhatele (2023) |
| 16 | "A large DNN model, trained on the same data, overcomes this overfitting by having access to many subnetworks or 'lottery tickets' " | Cowley et al. (2023) |
| 17 | "This may be due to the fact that having redundant parameters from the beginning of the training may make the loss landscape easier to optimize [139]; or it may be related to the increase in the likelihood of obtaining a "lottery ticket" [67]." | Kim et al. (2023) |

*Table 2.* Table 1 continued

| | Quote | Paper |
|---|---|---|
| 18 | "We show, theoretically and experimentally, that sparse initialization and increasing network width yield significant improvements in sample efficiency in this setting. Here, width plays the role of parallel search: it amplifies the probability of finding "lottery ticket" neurons, which learn sparse features more sample-efficiently." | Edelman et al. (2023) |
| 19 | "there exists a sparse subnetwork (winning ticket) that can be trained from scratch [...] In this view, a large model has a greater chance of containing a good subnetwork." | Lê et al. (2023) |
| 20 | "Frankle and Carbin [15] further conjecture that overparameterization improves performance because larger models contain exponentially more sparse subnetworks in superposition and are thus more likely to contain a "winning ticket" – a hypothesis supported by subsequent empirical and theoretical work [46, 44, 42, 3]." | Li et al. (2024) |
| 21 | "It also resonates with the Lottery Ticket Hypothesis, which proposes that successful training depends on the presence of a small sub-network—"the winning ticket"—within a larger model." | Díaz-Faloh & Mulet (2025) |
| 22 | "Evidence from the original work, supported by subsequent empirical (e.g. Zhou et al., 2019) and theoretical (e.g. Malach et al., 2020) studies, strongly indicates that the phenomenon exists broadly and the hypothesis holds." | Jeffares & van der Schaar (2025) |
| 23 | "While directly identifying lottery tickets before training remains impractical, the underlying ideas have significantly influenced our explanatory theories of sparsity, pruning, and network efficiency." | Jeffares & van der Schaar (2025) |

| | Quote | Online article or slides |
|---|---|---|
| 24 | "Since fat networks have exponentially more component subnetworks, starting from a fatter network increases the effective number of lottery tickets, thereby increasing the chances of containing a winning ticket. According to this hypothesis, pruning effectively identifies the subcomponent which is the winning ticket." | Huszár (2018) |
| 25 | "over-parametrization is not necessary for successful training - it may only help by providing a combinatorial explosion of available subnetworks" | Lange (2020) |
| 26 | "When the network is randomly initialized, there is a sub-network that is already decent at the task. Then, when training happens, that sub-network is reinforced and all other sub-networks are dampened so as to not interfere.[2]" | Kokotajlo (2020) |
| 27 | "If you want to win the lottery, just buy a lot of tickets and some will likely win. Buying a lot of tickets = having an overparameterized neural network for your task." | Princeton-CS-598D (2020) |
| 28 | "The SGD training process then solves the equations - it picks out the lottery tickets which perfectly match the data. In practice, there will be many such lottery tickets - many solutions to the equations - because modern nets are extremely overparameterized. SGD effectively picks one of them at random" | Wentworth (2021) |
| 29 | "Using the lottery ticket hypothesis, we can now easily explain the observation that large neural networks are more performant than small ones, but that we can still prune them after training without much of a loss in performance. A larger network just contains more different subnetworks with randomly initialized weights." | Fritz AI Blog (2023) |
| 30 | "The term derived from considering the probability of a tunable subnetwork as the equivalent to a winning lottery ticket; the chance of any given ticket winning is tiny, but if you buy enough of them you are certain to win, and the number of possible subnetworks increases exponentially as the power set of the set of connections, making the number of possible subnetworks astronomical for any reasonably large network." | Wikipedia (2024) |
| 31 | "It's like betting it all on lucky 13 a thousand times in parallel. You'd have to beat one-in-a-trillion odds not to win in that case. That's how training works. [...] This is the fundamental trick that makes artificial neural networks possible. This is how we cheat at the game. Combinatorics and graph theory and subnetworks and big, big, just stupidly big numbers." | Pondsiders (2025) |
| 32 | "The lottery ticket hypothesis crystallised: large networks succeed not by learning complex solutions, but by providing more opportunities to find simple ones. Every subset of weights represents a different lottery ticket—a potential elegant solution with random initialisation. Most tickets lose, but with billions of tickets, winning becomes inevitable." | Lord (2025) |

## A. Evidence of the problematic LTH interpretation

To get an unbiased sense of how widespread the interpretation in ref is, we employed two types of searches on the web: 1) systematic search on arXiv, and 2) search through LLMs standard or "deep research" abilities. Both were followed by human verification of the results. The results of both searches are reported in Tables 1 and 2.

### A.1. Systematic arXiv search

We queried Semantic Scholar for papers citing the original Lottery Ticket Hypothesis paper (Frankle & Carbin, 2018). The exact query is shown in Fig. Code A.1. As of 13 Nov 2025, the search returned 3,788 papers, out of which 2,503 were found to be linking to a version on arXiv. After retrieving the arxiv IDs and their latest version numbers (e.g., v3 in 2511.08092v3), we downloaded the PDFs of all papers from the `arXiv Dataset` available on Kaggle and subsequently transformed them to text using `pdftotext`. Using a 100-word sliding window, we selected word-blocks that included at least one word related to each of the three concepts:

- **Overparameterization**: ["overparameterized", "overparameterised", "large model", "wide", "big network", "overparameterised", "bigger", "fatter", "larger"]

- **Sub-network**: ["subnetwork", "lottery ticket", "subset", "path", "winning ticket", "subnetworks", "tickets", "ticket"]

- **Success**: ["initialization", "successfully", "train", "optimize", "reach accuracy", "succeed", "solution", "perform"]

This search produced a total of 3049 text blocks from 931 unique papers. With this rudimentary search method we aimed to maximize recall at the cost of precision. We further filtered these sentences by querying an LLM on filtering out false positives. The LLM used for this task (ChatGPT 5.1) was prompted to summarize its decision making, its response is reported in Box A.1. This filtering procedure produced 188 sentences. Finally, we manually verified that the quotes where kept verbatim and manually selected only the sentences that used the misleading elements of the analogy to explain the success of overparameterization, as described in Section 2.2. Other examples were added manually as the authors found them in the literature. The full list, is shown in Tables 1 and 2.

```
for i in 1:4
  https://api.semanticscholar.org/graph/v1/paper/[lth_paper_id]/
    citations?fields=title,authors,year,venue,externalIds,url&limit=[1000*i]
end
```

*Code A.1. Semantic Scholar query*

---

**Box A.1. LLM reporting its filtering strategy**

```
I filtered sentences by keeping only those that (1) explicitly mention
overparameterization or large/wide networks, (2) explicitly mention subnetworks or
lottery-ticket concepts, and (3) explicitly make the causal link that bigger networks
work because having many subnetworks increases the chance of finding a good one.
```

---

### A.2. LLM research results

We prompted ChatGPT-5.1 with searching for online sources, including articles, blog posts, and even slide decks, that interpret the lottery ticket hypothesis as suggesting that overparameterized networks work well because they contain many subnetworks, increasing the chances of having a winning subnetwork at initialization. We also asked the model to find such sources in "deep research" mode, where the model can search the web more extensively, see Box A.2 for the LLM self-reporting its strategy. While some results were satisfactory (albeit with many false positives), after manual verification, the amount of sources found was limited to around a dozen. That is why we decided to gather more sources by performing a systematic search of arXiv citations (described above).

---

**Box A.2. LLM reporting its deep research goal**

```
Each sentence must be from a source published between 2019 and 2025 and not authored
by Jonathan Frankle or Michael Carbin.  It must explicitly link overparameterization
(e.g., large, wide, or deep networks) to an increased probability of success in
training, via the presence of many candidate subnetworks or well-initialized
components.  The sentence should invoke the lottery ticket metaphor, describing how
large models "buy more tickets" or embed more trainable subnetworks.  Sentences that
merely define the lottery ticket hypothesis or discuss pruning without making this
probabilistic-mechanistic connection should be excluded.
```

## B. Simulation methods

### B.1. Fig. 3 *top*

We train a two-layer neural network on a synthetic task to investigate whether subnetworks that successfully learn the task in isolation can be made to fail by adding additional random neurons to the network.

**Data and teacher network:** We use a synthetic regression task where the target function is defined by the teacher:

$$f_{\text{teacher}}(\mathbf{x}) = \text{ReLU}(\mathbf{w}_1^T\mathbf{x} - b) + \text{ReLU}(\mathbf{w}_2^T\mathbf{x} - b) + \text{ReLU}(\mathbf{w}_3^T\mathbf{x} + b) - \text{ReLU}(\mathbf{w}_3^T\mathbf{x} - b) \tag{8}$$

where $\mathbf{x} \in \mathbb{R}^2$, $\mathbf{w}_1 = [1,1]$, $\mathbf{w}_2 = -\mathbf{w}_1$, $\mathbf{w}_3 = [1,-1]$, $b = \sqrt{3}/2$. Input data consists of 30,000 samples drawn uniformly from $[-\sqrt{3}, \sqrt{3}] \times [-\sqrt{3}, \sqrt{3}]$ (unit variance distribution).

**Student networks architecture:** A two-layer network with $r \in \{4, 5, 6, 7\}$ ReLU hidden units and one output unit. Both layers have bias terms.

**Training procedure:** We train student networks from random initialization with different random seeds. Since we are in a teacher-student setup where teacher and student have the same architecture, there exist parameters that achieve zero loss. We consider a training successful if the dynamics of gradient descent lead to a global minimum, i.e. final loss is below $10^{-25}$. Note that in over-specified students, there exist multiple global minima that achieve zero loss (Simsek et al., 2021). Training is performed with the MLPGradientFlow package (Brea et al., 2023) using the differential equation solver KenCarp58. With this high precision procedure, we make sure that gradient descent has converged, without incurring into the risk of mistaking convergence for the traversal of flat regions of the landscape. We run the ODE solver for a maximum of 15 seconds and apply a second-order optimizer for 1 additional second to ensure convergence to local minima. We make sure that with this procedure, all trained networks have converged to a local minimum by verifying that for at least $10^5$ iterations the loss does not decrease further.

**Winning tickets identification:** From the 100 baseline trainings, we identify 7 networks that successfully converged to small loss. These trained networks serve as our "winning tickets".

**Adding neurons experiment:** For each winning ticket (identified by its initialization $I^*$), we test whether the subnetwork remains a winning ticket when embedded in a larger network ($r \in \{5, 6, 7\}$). Specifically, we:

1. Initialize the first 4 hidden neurons from the winning ticket, including weights and biases from both layers

2. Randomly initialize $r - 4$ new hidden neurons using Glorot normal initialization with 30 different seeds

3. Concatenate the ticket parameters with the new random parameters, resulting in a network of $r$ hidden neurons

4. Train this network for the same convergence criteria as before

We perform this experiment for each of the identified winning tickets and each of the 30 random seeds, resulting in multiple trials per ticket. We record the final loss value to determine whether the embedded ticket still learns the task successfully.

**Random re-initialization** As a control experiment, we also test the base success rate of each network size. We follow exactly the same procedure as above, but we randomly initialize the parameters from a Glorot normal distribution.

**B.2. Fig. 3** *bottom*

We empirically test the prediction that the probability of training failure decreases exponentially with the number of subnetworks embedded in a dense network, as suggested by the independence assumption in Eq. 4.

**Data and teacher network:** We consider the same teacher–student regression task as above. The teacher is a two-layer MLP with 4 hidden neurons and no output bias,

$$f_{\text{teacher}}(\mathbf{x}) = \sum_{i=1}^{4} \sigma(\mathbf{w}_i^\top \mathbf{x}),$$

with fixed weights $\mathbf{w}_i \in \mathbb{R}^2$ given by $\{(0.6, -0.5), (-0.2, 0.1), (0.5, 0.5), (-0.6, -0.6)\}$. The dataset is generated by evaluating the teacher on 1600 input samples drawn from the same distribution as in the other experiments. Since the student and teacher share the same architecture, zero training loss is achievable with width 4.

**Student networks architecture:** Student networks are two-layer MLPs with a single output neuron and biases in both layers. We vary the number of hidden neurons $r \in \{4, 5, 6, 7\}$.

**Training procedure and success criterion:** All networks are trained from random initialization using the same high-precision gradient flow procedure as in the top experiment. A training run is declared successful if the final loss is below $10^{-25}$, ensuring convergence to a global minimum. As before, we verify convergence by checking that the loss does not decrease over at least $10^5$ iterations. To obtain a precise estimate of failure probabilities, especially when small, we train 4000 differently initialized students per value of width.

**B.3. Fig. 5**

We describe the numerical experiment used to illustrate how neuron addition can create escape directions in a nonconvex loss landscape.

**Data and teacher network:** We consider a synthetic teacher–student regression task. The teacher is a two-layer MLP with Softplus activations and 4 hidden neurons, defined as

$$f_{\text{teacher}}(\mathbf{x}) = \frac{1}{2}\,\sigma(\mathbf{w}_1^\top \mathbf{x} - b) + \sigma(\mathbf{w}_2^\top \mathbf{x} - b) + \sigma(\mathbf{w}_3^\top \mathbf{x} + b) - \frac{3}{2}\,\sigma(\mathbf{w}_4^\top \mathbf{x} - b),$$

where $\sigma(\cdot)$ denotes the Softplus function,

$$\mathbf{w}_1 = (1, 1), \quad \mathbf{w}_2 = (-1, \tfrac{1}{3}), \quad \mathbf{w}_3 = (\tfrac{1}{3}, -1), \quad \mathbf{w}_4 = (-\tfrac{1}{2}, -1),$$

and $b = \sqrt{3}/2$. Input samples $\mathbf{x} \in \mathbb{R}^2$ are drawn from the same distribution as in the other experiments.

**Student networks architecture:** The student networks are two-layer softplus MLPs with a single output neuron and bias terms in both layers. We consider:

- a *small network* with 3 hidden neurons, and

- a *large network* with 4 hidden neurons obtained by splitting one neuron of the small network.

Both networks use identical architectures apart from the number of hidden units.

**Training procedure:** The small network is trained from random initialization using gradient descent until convergence to a local minimum. Training minimizes the mean squared error loss. We ensure convergence by running the optimization until the loss ceases to decrease further. All training is performed using the same numerical integration and convergence criteria as in the other experiments.

**Neuron splitting operation:** We experimented with both $\epsilon = 10^{-15}$ and $\epsilon = 0$ and obtained similar results. We speculate that $\epsilon = 0$ still works due to machine precision error (the two neurons, if identical up to infinite precision should never split).

