# OpenReview forum: "Position: Lottery Tickets Are Misleading! Use Escape Dimensions to Explain the Success of Overparameterization"
_ICML.cc/2026/Position_Paper_Track — Submitted to ICML 2026 Position Paper Track_

### Official Review · Reviewer_qShV · 2026-03-06

**Significance:** 4
**Argument Clarity:** 4
**Ethics Flag:** Yes
**Rating:** 6
**Confidence:** 5

**Questions:**

No questions

**Alternative Views Section:**

Yes

**Compliance With Llm Reviewing Policy A Conservative:**

Affirmed.

**Discussion Potential:**

4

**Final Justification:**

I still believe in acceptance of this paper; the rebuttal did not change my score.

Overall, I believe that even if escape dimensions are not the final explanation for the phenomenon (other reviewers seem to not be convinced by this point), this paper to me is the ideal of what a position track paper should be; take some idea that has a high acceptance rate in the community, really dig into the details, and provide some alternative viewpoints. Therefore I strongly recommend acceptance to the position track.

**Paper Summary:**

This paper examines the "lottery ticket hypothesis" that networks contain subnetworks which, at initialization, are sufficient to recapitulate the training performance of the full network. They detail the various versions of the argument, and then construct a simple experiment with MLPs that violates the principles of the hypothesis. The authors then present an alternative viewpoint, which suggests that simple descriptions of high dimensional loss landscapes and gradient descent dynamics are sufficient to explain the phenomena that the lottery ticket hypothesis is trying to address.

**Position:**

Yes

**Position In Title:**

Yes

**Related Work:**

4

**Strengths And Weaknesses:**

The main strengths of the paper is that it is incredibly well written, and detailed in its references to the literature. In addition, the simple experiment with the MLP is a very good demonstration of the basic principles, and is easy to follow.

The main weakness is that the alternative arguments section could present better alternative arguments. One argument would be against the MLP experiment itself; there is some argument that what happens in low dimensional settings is not indicative of what happens in high dimensions. There is also a broader argument that escape directions is not the right way to describe the phenomena at play; see for example works like [this](https://proceedings.mlr.press/v97/du19c.html).

**Support:**

4

---

> ### Author Rebuttal · Authors · 2026-03-31
>
> We thank the reviewer for the constructive and positive feedback. We appreciate that the reviewer liked our exposition style. Finding a good structure to present our position was a very non-trivial exercise.
>
> ## Alternative views
>
> We agree that the alternative views section was too short, and expanded it considerably in the updated manuscript. In particular, we developed the discussion around the neural race reduction (Saxe 2022) with new analytical evidence from Pinson 2026, who derives equations for the dynamics of neurons under strong assumption (yet to be seen if they could be relaxed). Pinson proposes a correction and refinement (race) rather than dismissing the lottery metaphor. Comparing to them, we acknowledged that our work addresses trainability only, leaving understanding dynamics, as well as generalization and implicit bias to future work (Wilson, 2025). We also mention that not all global minima in overparameterized landscape generalize well, but for this we refer to discussions about rich and lazy regimes (Chizat et al. 2019).
>
> ## Low- vs. High-Dimensional Setting
>
> In general, we agree that one should be skeptical about transferring insights from low dimensions to high dimensions. For example, the local minima on lines of saddle points, i.e. plateau saddles (Fukumizu and Amari, 2000) are unlikely to occur in high dimensions (Petzka and Sminchisescu, 2021). However, escape dimensions allow escaping from higher losses to lower losses in any dimension. Hence, we see no reason why the generalization should fail in this case. The low-dimensional simulations are meant as didactic examples to convey the intuition.
>
> With standard initialization, SGD may not pass nearby saddle points. For example, the trajectory in figure 5, which almost approaches the saddle point, may be unlikely with standard initialization. However, the escape dimensions exist nevertheless, and prevent SGD from getting stuck at high losses. This is the mechanism that transforms the geometry of the landscape. For very wide neural networks, we agree with the view of Du et al. 2019 (thanks for the reference, we will include it in the citations): the global loss manifold dominates the loss landscape (at least in regions of standard initialization) as illustrated in the sketch in the last figure (right most panel). The reason for this dominance of the global loss manifold is that _all higher-loss local minima disappeared due to adding escape dimenions_ ([see fig.3 in the linked pdf](https://mega.nz/file/T2Q2nDyR#n5gTJPW80TTuRPVKlOM81x-bCjJhP8Z4fi3DIWoUGQ0)).
>
> On a more practical side, to address the concern about low-dimensional settings, we designed an experiment on Frankle & Carbin's Conv-6 architecture trained on CIFAR-10. We called this adversarial Overparameterization (advOP). Starting from lottery tickets found by iterative magnitude pruning, we regrow a fraction of pruned connections and initialize them to maximally oppose the ticket's gradient signal. The winning ticket is now embedded in a larger network, but the rest of the connections concoct adversarially. Results in the [linked figures](https://mega.nz/file/T2Q2nDyR#n5gTJPW80TTuRPVKlOM81x-bCjJhP8Z4fi3DIWoUGQ0) show that advOP consistently degrades the ticket's performance across sparsity levels, confirming that subnetwork outcomes depend on the rest of the network even in a realistic, high-dimensional setting.

---

> > ### Author Rebuttal · Reviewer_qShV · 2026-04-02
> >
> > Thanks for the responses; I will keep my score.

---

### Official Review · Reviewer_kEUX · 2026-03-09

**Significance:** 3
**Argument Clarity:** 2
**Rating:** 2
**Confidence:** 4

**Questions:**

*Q1* Is Escape Dimensions Theory a term already in use? If so, by which of the cited references in the paper? If not, is it necessary to create a new term?

*Q2* Why not make the position be about using loss landscapes as a more comprehensive and intuitive explanation of why training succeeds in the overparameterized regime?

*Q3* Are the statements for independence, sufficiency, and scaling associated with LTH  made after the original LTH paper or based on the distortions produced ever since?

*Q4* Please address my comments regarding optimization.

*Q5* Somewhat related to Q3: what backs the sufficiency claim stating that "if there exists at least one winning ticket at initialization, then the dense network will succeed" (Page 4) and "[t]he presence of the winning ticket inside the network would predict a success rate of 1 (Page 5, Figure 3)?

*Q6* Ending Section 3 by mentioning "explicit and implicit assumptions" is not very reassuring. Can you please have a list of what those are?

*Q7* Somewhat related to Q3: when you discuss "assumptions of lottery tickets" at the end of Section 3, are you acknowledging that those are not a direct consequence of LTH as originally stated, but rather the result of how LTH was distorted over time?

*Q8* Please address other issues regarding Section 3 from my assessment of strengths and weaknesses.

*Q9* Isn't going from 3 to 4 neurons in the experiment in Section 4 too benevolent to your claims, given that the teacher network has 4 neurons, whereas you go for slightly larger networks in the experiment in Section 3?

*Q10* It neuron splitting either necessary or sufficient for training success in the overparameterized regime?

*Q11* Given that neuron splitting depends on certain conditions being met, can it suffice to explain EDT? For example, does it justify the the argument backed by Simsek et al. (2021) in Section 4.2?

*Q12* Please address other issues regarding Section 4 from my assessment of strengths and weaknesses.

*Q13* I would like your thoughts regarding the sufficiency of the Alternative Views section.

*Q14* Aren't there other explanations for why subnetworks are as successful as the entire network?

*Q15* Isn't LTH used more often in the context of justifying network pruning than in the context of justifying success in overparameterized training?

**Alternative Views Section:**

No

**Compliance With Llm Reviewing Policy A Conservative:**

Affirmed.

**Discussion Potential:**

2

**Final Justification:**

I remain concerned with the strawman arguments used in this paper, along with the sensationalistic title, for which reason I will keep my score.

**Paper Summary:**

This position paper objects to the use of the Lottery Ticket Hypothesis (LTH) for explaining the success of training neural networks in overparameterized regimes. While not denying the validity of the claims made by Frankle & Carbin in the paper that originated the LTH as its accompanying Lottery Ticket Conjecture, the authors argue that the intuition behind the LTH limits our understanding of what happens more broadly in the overparameterized regime, and also that the original claim and conjecture have been reinterpreted in ways that have not preserved its intended meaning - in some cases implying something that has never been formally proven. The authors propose an alternative perspective based on what happens to the loss function landscape (or, simply, loss landscape) as the neural network grows larger. They denote this alternative perspective as Escape Dimensions Theory (EDT).

**Position:**

Yes

**Position In Title:**

Yes

**Related Work:**

2

**Strengths And Weaknesses:**

## Strengths

- The first illustration of how EDT may validate the intuition behind loss landscapes in higher dimensions (Figure 1) is very helpful and intuitive.

- The collection of quotes indicating how the LTH has been (mis)characterized in the literature and other materials backs one of the main concerns expressed by the authors. I also liked how they created a compact way of citing those quotes along the text.

- The parallel between a conventional lottery and the LTH interpretation of lotteries helps pointing out the subtle difference between the two, which is often overseen in some of the claims about training that are made with LTH in mind.

- Pointing out that Strong LTH has nothing to do with training and moves away from understanding successful training in the first place is important and helpful for the discussion.

## Weaknesses

- It is not clear from the paper if Escape Dimensions Theory is a term already coined and adopted in the literature, since the box in the first page credits no one. If coining this term is part of the paper contribution, I believe that creating a new term should be justified - especially considering that this is a position piece. Consider the cons: proposing a theory may produce the same vices as LTH for becoming a static explanation or, worse, one which gets progressively distorted over time - as the authors have argued about LTH. Out of concern for not de-anonymizing the submission, I decided not to search for this term. See Questions 1 and 2.

- Although I like the parallel between a conventional lottery and the LTH interpretation of lotteries, as pointed out in the strengths, it is not entirely clear if the statements for independence, sufficiency, and scaling are made after the original LTH paper or based on the distortions produced ever since. Again, there are no explicit citations for any of the statements. See Question 3.

- While discussing Strong LTH, the authors miss addressing how this have been used in papers about pruning at initialization.

- The way by which optimization is discussed in the paper lacks proper intuition and is sometimes inaccurate (See Question 4):
  - I do not understand what the authors mean by "[t]raining no longer plays a role in optimizing a solution" (Page 4). What follows this sentence if a possible interpretation for what happens during optimization. Moreover, at least for the subnetwork, what optimization does follows the common intuition for it.
  - The rationale implied by Equation 1 voids optimization of any role. Taking it at its face value, any optimization algorithm should have the same result. The same would be true with EDT in Section 4, although the authors do not push EDT to the same extreme to make it evident.
  - The argument about subnetwork independence (Page 5) is not as relevant if you consider that in some lotteries you may have tickets with overlapping numbers getting partial rewards. It is not clear why independence seems to be a problem here.
  - Training neural networks is NP--hard in general (with an en-dash, not a hyphen). While that is likely due to the overparameterized regime, this is an assumption of the authors that is not backed by the citation used in Page 6.

- Up to Section 2, I was leaning toward acceptance with some reservations. However, I believe that Section 3 went too far, and in doing so it discredited the position (see Questions 5 to 8):
  - Equation 1 pushes LTH to such an extreme that it becomes a strawman argument.
  - The experiment following Equation 1 is clearly not under the overparameterized regime, so it does not address the claim made in the title of the position paper.
  - The argument about the gradient of the hidden layer being "dependent on the entirety of the larger network" misses the point that the original LTH paper uses magnitude prunning, and therefore removes connections because they have a small absolute value. Unfortunately, pruning is only really discussed in some depth by the end of Section 4.3 in Page 8.

- Conversely, Section 4 is excessively benevolent in favor of the position stated (see Questions 9 to 12)
  - The authors show the benefit of a larger network by adding a fourth neuron to learn from a teacher network that has four neurons. For coherence, shouldn't they be training in larger networks than the original, as in Section 3?
  - It is not clear if neuron splitting is either stated necessary or sufficient to allow training to succeed in larger networks. Moreover, it does not seem to be the case that neuron splitting can always justify success in wider networks, given that it only happens under certain conditions.
  - The argument about winning tickets in Section 4.3 is not about the overparameterized regime, so it is again irrelevant for the position in the title of the paper.
  - It is not clear how subnetwork independence plays a role in the argument of Section 4.3, especially because the sentence does not explain why.

The Alternative Views section only has one paragraph honoring that purpose. That paragraph presents only two sentences about what other perspectives exist, and in talking about activated subnetworks it ignores a broader related literature on linear regions, which have also been used for orienting network pruning. It could be argued that the opposing view is what the authors have described earlier in the paper about LTH itself, but the presentation elsewhere in the paper always has a critic undertone, for which reason it does not justify the absence of a proper Alternative Views section. On the other hand, the authors do a fair self-assessment of their argument in pointing out that they have not addressed generalization, benign overfitting, and implicit bias. See Questions 13 and 14.

It is also left unaddressed the fact that LTH was meant to justify the success of pruning, not of overparameterized training. See Question 15.

### Minor issues
- Page 1, Introduction, remove comma in "make sense of this complexity, by stripping down"
- Page 2, Section 2, remove comma in "was surprising, because it seemed"
- Page 2, Section 2, remove comma in "a large body of work, with the dream of"
- Page 2, Section 2, clarify that Tables 1 and 2 can be found in appendix A
- Page 2, Section 2, replace "would be" with "were" in "the lottery analogy wold be interpreted literally"
- Page 4, Section 2.3, replace "pre-exist" with "exist before" (or equivalent)
- Page 4, Section 3, these "two-layer" MLPs have only one layer
- Page 4, Section 3, "[b]ecause of the random initialization, the pool of potential subnetworks is infinite" does not apply finite precision, such as with floating-point arithmetic
- Page 5, Second last paragraph, remove comma following partial derivative in "the original subnetwork [partial derivative], is not dependent"
- Page 7, Figure 5, it is not easy to understand that the teacher weights are connected to a central point consisting of the origin and I did not see a reason for connecting those weights to the origin if that is not explicitly discussed; consider removing it
- Page 7, Bottom of left column, strange use of "three hidden neurons": what is not a "hidden" neuron?
- Page 8, Section 5, remove comma in "is misleading, because it"
- Appendix, Tables and 1, place those tables as part of Appendix A rather than thrown before them
- References, Why cite the arXiv version and not the ICLR 2019 publication of the LTH paper?

**Support:**

3

---

> ### Author Rebuttal · Authors · 2026-03-31
>
> We thank the reviewer for the depth of this review. We try to address it within the 5k limit but encourage the reviewer to engage further if any point deserves closer attention.
> If our responses helped clarify the sources of disagreement, we kindly ask the reviewer to reconsider their rating.
>
> *In italics we indicate the actions we took to modify the draft.*
>
> ### Escape dimensions
>
> **Q1**.We coined this term (*will acknowledge in the text*) to name a specific mechanism within loss landscape theory, grounded in theorems (unlike the LTC). The mechanism has been known since Fukumizu and Amari, 2000 but remains overlooked; we believe a name helps it gain the visibility it deserves.
> If future work proves it incomplete, we welcome that.
>
> **Q2**. Our position is precise: that escape dimensions is a better mental model than lottery tickets to explain the success of overparameterization.
>
> **Q10/11/12.2**. The result of Fukumizu and Amari (2000) requires no assumption beyond having an architecture made of units (neurons, channels, etc.): every neuron addition transforms minima into saddles. One does not have to choose the new neuron's parameters, just adding new dimensions changes the stability of critical points. The results of Simsek built on this.
> As stated in the paper, neuron splitting is the mechanism enabling gradient descent to find new, lower loss minima. It shows that minima inherited from smaller networks are no longer traps, because they become saddles with escape dimensions. As we state in the discussion, ED may not be the only actor at play, but remains a solid mental model.
> *We updated the text to make the generality of the theorem clear.*
>
> ### Strawman (Q3, Q4.1, Q4.2, Q5.1, Q6, Q7, Eq.1).
>
> We respectfully disagree that our formalization is a strawman. The 30+ quotes directly invoke sufficiency and scaling, and indirectly invoke independence (to justify the emergence of combinatorial gain of width, quoting wikipedia). These properties are inherent to any literal lottery, and their translation to subnetworks is suggested in the LTH original paper (see response point 2/3 to `dsSd`) and reinforced by community usage.
> The quotes rarely commit to a precise mathematical claim; this vagueness is itself the problem. Our choice is to concretize these statements by taking the analogy literally, then show the resulting predictions fail. This argumentation strategy is sometimes called **reductio ad absurdum**.
> *We will clarify the origin of these three properties and their translation to subnetworks by citing the relevant quotes from the LTH, and stressing that we are taking the metaphor literally.*
>
> **Q6**: *this sentence was vague and added no value, we removed it from the text*, we referred (confusingly) to sufficiency and scaling vs. implicit independence.
>
> The reviewer is correct that Eq. 1 voids optimization of any role: this is precisely what makes the lottery analogy misleading, and why we argue against it!
> Eq. 1 is a direct formalization of what a lottery means: if a winning ticket is present, you win. Buying another one or another thousand will not affect the outcome (even with overlapping rewards). This is what the Lottery Ticket Conjecture and subsequent quotes reduce to: the role of SGD is not to train all weights by shaping a solution, but to seek out a winning ticket and train it in isolation (section 2.3). This view is first put forward by the original paper (see response 2/3 to `dsSd`) and reinforced by multiple quotes.
> EDT is fundamentally different: minima becoming saddles is only useful because gradient descent can exploit escape dimensions.
>
> ### Overparameterization
>
> **Q5.2**. The lottery analogy suggests that having a winning ticket embedded in a network's initialization guarantees successful training (hence the predicted success rate of 1 in Fig. 3).
>
> **Q8.2/12.3**. A 5-neuron student learning a 4-neuron teacher **is overparameterized** because the student has more capacity than necessary to learn the dataset, consistent with the standard definition in the theoretical literature (e.g., Belkin, 2021, Simsek et al. 2021).
> *We will explicitly state that experiments of Fig. 3 are overparameterized and why*
>
> **Q8.3**. The ticket's gradient depends on the rest of the network throughout training. Magnitude pruning removes connections close to zero **at the end of training**, but these connections are not zero at initialization and influence the ticket's dynamics.
>
> **Q9/12.1**. the reviewer is right: for consistency it is better to show escape dimensions effects going from 4 to 5 neurons. *[here's the new fig.5](https://mega.nz/file/T2Q2nDyR#n5gTJPW80TTuRPVKlOM81x-bCjJhP8Z4fi3DIWoUGQ0)*
>
> ### Remaining points.
>
> Q13: **We acknowledge they were far from sufficient!** See response to reviewer `qShV`.
>
> Q14: this question is beyond the scope of our position
>
> Q15: we are not discussing anything else than the use of LTH to explain overparameterization
>
> Minor corrections: incorporated. Thank you!

---

> > ### Author Rebuttal · Reviewer_kEUX · 2026-04-02
> >
> > The abuse of equation (1) remains a strawman argument.
> >
> > There is no fig. 5 in the linked document.
> >
> > If you are discussing LTH in a very specific context, your paper title should not start with "Lottery Tickets Are Misleading!" without any qualifier before the exclamation mark. Throughout the paper, you leave no room for benevolently acknowledging LTH for its contributions. Scholarship is a constructive work.
> >
> > In coining a term, especially if based on third-party results, I believe that it would be meritorious to first cite those authors on page 1 and next to the framing instead of under section 4.

---

### Official Review · Reviewer_n2bA · 2026-03-11

**Significance:** 3
**Argument Clarity:** 3
**Rating:** 5
**Confidence:** 3

**Questions:**

1. To support the claim that the scaling property does not generally hold, wouldn't neural scaling laws provide a natural counterexample? Empirically, the model performance follows a power-law trend with scale, rather than the kind of exponential behavior suggested by LTH.

2. A related but slightly digressive question, could the notion of escape dimension shed any light on neural scaling laws?

**Alternative Views Section:**

Yes

**Compliance With Llm Reviewing Policy A Conservative:**

Affirmed.

**Discussion Potential:**

2

**Final Justification:**

My concerns are addressed so I keep my positive evaluation.

**Paper Summary:**

This paper argues that using the lottery ticket hypothesis as an analogy for the success of overparameterized models is misleading. Specifically, the authors first show how existing literature often assumes three properties to subnetworks embedded in larger networks, namely independence, sufficiency, and scaling, and collect representative statements that reflect these claims. They then use toy experiments on small two-layer neural networks to show that none of these properties really hold for subnetworks inside an overparameterized model.
Based on these observations, the paper argues that overparameterization should be understood through learning dynamics and training trajectories, rather than through the existence of a good subnetwork initialization. In particular, the authors propose the notion of an escape dimension as a way to explain the nide optimization behavior of overparameterized models, and support this intuition with additional toy experiments. Overall, the authors call for moving away from lottery ticket analogy (which they argue create an unhelpful mental picture for the community) and instead focusing on optimization geometry and the loss landscape to explain why overparameterization works.

**Position:**

Yes

**Position In Title:**

Yes

**Related Work:**

4

**Strengths And Weaknesses:**

**Strengths.**

1. The opposite view against the lottery ticket analogy (especially the claimed independence, sufficiency, and scaling properties) is well supported by clear and informative toy experiments.

2. The proposed use of the escape dimension as an explanatory lens for optimization in overparameterized models is insightful, and the empirical results make the intuition easy to follow. This direction seems promising and worth further study.

3. The paper provides a helpful survey of potentially misleading statements in previous work that apply the lottery ticket hypothesis to explain overparameterization, and it also lists many relevant perspectives from the loss landscape literature.

**Weaknesses.**

1. While I appreciate the evidence that the three lottery ticket properties may not hold for subnetworks within an overparameterized model, I am not fully convinced that this makes the lottery ticket analogy broadly misleading. The lottery ticket (or the existence) viewpoint and the loss landscape viewpoint can be seen as studying different facets of theory of overparameterization, roughly similar to expressiveness versus optimization within the  standard aspects of expressiveness, optimization, and generalization in deep learning theory. Claiming one as misleading and solely emphasizing the importance of the other may be overstated. They could be complementary rather than competing explanations.

2. My another concern is that the paper's main call, namely to focus on learning dynamics and loss landscape structure when explaining deep learning's success, may be less novel than suggested. I think the community has increasingly recognized that gradient-based training dynamics are central not only for optimization but also for generalization. The paper indeed collects many examples where trajectory-based explanations are used to study overparameterization.

**Support:**

4

---

> ### Author Rebuttal · Authors · 2026-03-31
>
> We thank the reviewer for their thoughtful comments. *In italics we indicate the actions we took to modify the draft.*
>
> ## Questions
>
> 1. This is a very interesting take! They indicate indeed a different scaling behavior than suggested by combinatorial arguments based on LTH. But for the sake of rigor, the observed scaling laws with model size depend on hyperparameter choices whose impact are arguably not yet fully understood. For the moment, we see this as speculative evidence, and *will mention in the discussion*.
> 2. Great question! Short answer: we do not know yet. Escape dimensions allow escaping from high-loss local minima to lower loss values. How much the loss decreases per escape dimension (in expectation) is currently an open question, with only indirect and partial answers in e.g. works inspired by statistical physics.
>
> ## Complementariness
>
> We think the Frankle & Carbin paper is great, in particular their empirical findings on pruning. We resonate with how you refer to the LTH: "The lottery ticket _(or the existence)_ viewpoint" (our emphasis). Indeed, the empirical statement in the Frankle & Carbin paper could be called "existence-of-trainable-sparse-subnetworks hypothesis" instead of "lottery ticket hypothesis". We totally agree that the LTH understood this way is a nice complementary to the landscape viewpoint. *We clarify this further in the introduction and discussion of the updated manuscript.*
> On the other hand, we think that the lottery metaphor and any predictive or explanatory claims too close to the rules of lotteries are misleading; if not backed by theoretical arguments that at the moment are lacking.
>
> ## Novelty
>
> We agree that there exists already an important body of work focusing on learning dynamics and loss landscape structures (see our citations). Our intention is to foster more research in this direction, while giving credit to seminal works on loss landscape structures (e.g. Fukumizu and Amari, 2000). *We clarify this further in the updated alternative views and discussion sections.*

---

> > ### Author Rebuttal · Reviewer_n2bA · 2026-04-02
> >
> > Thank you for your response. I have no further questions, and I would like to maintain my evaluation that this paper is a clear accept.

---

### Official Review · Reviewer_dsSd · 2026-03-16

**Significance:** 3
**Argument Clarity:** 1
**Rating:** 2
**Confidence:** 3

**Questions:**

N/a

**Alternative Views Section:**

No

**Compliance With Llm Reviewing Policy A Conservative:**

Affirmed.

**Discussion Potential:**

3

**Final Justification:**

I'd like to thank the authors for clarifications of my previous post. I do believe that clear separation between LTH and LTC is important and the authors have a point separating those. At the same time I still hold my position that most of the analysis carried out be the authors is based on the artificial setup that the authors have derived assuming literal interpretation of the "lottery". Please notice that authors have clearly stated this in the response to the reviewer kEUX:

> Our choice is to concretize these statements by taking the analogy [of lottery tickets] literally, then show the resulting predictions fail.

But that's **authors choice** to take the analogy literally and thus I believe that's what the reviewer kEUX meant when referring to the strawman argument.

At a more analytical level, I'd like to mention one discrepancy between LTH and authors sufficiency claim. Notice that when LTH defines a winning ticket it's a subnetwork **matching the performance of the dense counterpart**. In the authors setup which claims to disprove the sufficiency condition, the success criterion is based on **matching the performance of the subnetwork**. This is clearly a different case, one can easily imagine three sets: $\mathcal{A}$ - set of all winning sparse networks, $\mathcal{B}$ - set of all networks trainable to accuracy $p$, $\mathcal{C}$ - set of all networks untrainable to accuracy $p$. Clearly, $\mathcal{C} \subset \mathcal{A} \cup \mathcal{B}$ and $\mathcal{A} \cap \mathcal{B} = \varnothing$. But most importantly, the split between $\mathcal{A}$ and $\mathcal{B}$ is derived based on the accuracies of dense models. It is not stated anywhere in the original work of LTH, but it is obvious that the authors are interested in understanding the relation between $\mathcal{A}$ and $\mathcal{C}$. While in this work, the authors have created a different setup showing that it is possible to find such an element of $c \in \mathcal{C}$ that $c \notin \mathcal{A}$. The problem with this argument is that it is trivial and not really interesting from the perspective of LTH, because the work does not claim anything about these cases (it focuses on the success cases from $\mathcal{A}$ only). To understand why the result is trivial, one can always follow a procedure where one chooses an element $c \in \mathcal{C}$ (remember that these subnetworks must achieve accuracy $p$ when trained in isolation) and augments its with arbitrary number of zeroed-out layers and neurons. Clearly, such model won't achieve accuracy $p$ due to the problems with signal propagation but according to the sufficiency condition states by the authors eq (1), since we have our winning ticket, we must be able to train the dense part as well. One can even define a case where no additional layers are created, but simply the weights in the same layer are chosen in a way that cancels out the signal form the winning ticket and would have equal problems with training this augmented network.


To summarize: I think the main problem of this work comes from treating the analogy to literally by the authors and deriving multiple flawed conclusions out of this analogy. Additionally, the rebuttal haven't increased the significance of the escape dimensions approach in any meaningful way thus, at the current form the work does not much value of the community. Therefore I keep my score unchanged.

**Paper Summary:**

The paper studies the idea of lottery ticket hypothesis (LTH) which claims the existence of sparse subnetworks which can be trained from randomly initialized state and reach commensurate accuracy to the dense counterpart. In particular, the paper states the position that the exact formulation of "lottery tickets" is misleading and has lead to multiple wrong conclusions. Additionally, the authors claim that the perspective of escape dimension is a more precise explanation of the empirical findings presented in the seminal LTH work.

**Position:**

Yes

**Position In Title:**

Yes

**Related Work:**

2

**Strengths And Weaknesses:**

## Strengths

1. The topic of LTH has gained a significant traction a couple years ago and remains a heavily investigated since then. Thus a work offering a different perspective on this topic would be interesting to a large part of the ICML's audience.

2. The overall structure and presentation clarity is top notch.

## Weaknesess

1. The main position is slightly unclear to me due to its dual nature. It is basically composed of two claims: 1) Lottery Tickets are Misleading. 2) Escape dimensions explain it better. While the authors do provide supportive arguments for both of these claims, I feel like the latter is treated additionally with rather vague explanations and I feel that focusing on a single claim would make this argument easier. Also, if the latter is also a part of the main position, then the Alternative View section entirely misses this point -- that's why I voted "No" in this case.

2. The whole work claims to challenge popular misconceptualization of the original LTH work rather than the work itself which is clearly stated by the authors of this work. This makes it a bit hard to argue with as this "misconceptualization" is quite freely defined by the authors through some quotes derived from other works and lectures, but just because the authors have found numerous examples of such statements it does not imply that this is a leading viewpoint of the community. Also, the authors use the same argumentation to form rather arbitrary statements of Independence, Sufficiency and Scalability of the tickets -- none of which is rooted original LTH.

3. These statements are then used to derive a flawed analytical argumentation against the LTH. To be more precise, according to the authors:

> The sufficiency property states that if there exists at least one winning ticket in the initialization, then the dense network will succeed:

But in reality, the original LTH states that:
> A randomly-initialized dense neural network contains a subnetwork that is initialized such that -- when trained in isolation -- **it can match the test accuracy of the original network** after training for at most the same number of iterations.

According to this formulation one clearly see that even networks having random-guess accuracies contain these subnetwork (i.e. lottery tickets). This makes the whole formulation wrong and since the whole argumentation in Section 3 is dervied from this assumption it makes all these arguments flawed.

4. I do appreciate that the authors tried to support both of their positions with some empirical experiments, however the proposed experiments are far from being convincing. This especially applies to the latter position (escape dimensions) which is supported in a hand-wavy way with some arbitrary illustrations of loss landscapes and a toy experiment illustrating authors claims -- much too little to be considered as a significant balance for the empirical results from the LTH work.

**Support:**

1

---

> ### Author Rebuttal · Authors · 2026-03-31
>
> We thank the reviewer for raising several important points. Below we will clarify each of them.  *In italics we indicate the actions we took to modify the draft.*
>
> We are afraid that two key messages were not conveyed clearly:
>
> First, the fundamental difference between the Lottery Ticket Hypothesis (LTH, the empirical finding) and the Conjecture (LTC, in the same paper).
>
> Second, the importance of mental pictures that develop a life of their own. Our position argues for a transition of mental pictures: "escape dimensions is a better mental model than lottery tickets to explain the success of overparameterization."
>
> ### We do not challenge nor explain the empirical findings of LTH
>
> From the reviewer summary
> > the authors claim that the perspective of escape dimension is a more precise explanation of the empirical findings presented in the seminal LTH work.
>
> This is not our claim, and may have led to the concerns in points 3 and 4. We do not question the existence of sparse trainable subnetworks, nor aim to explain them.
> We challenge the lottery analogy as a mechanism for why overparameterization works (the LTC), which has no evidence supporting it.
>
> The reviewer also argues that "even networks having random-guess accuracies contain these subnetworks," referring to the strong LTH. Ref. to Section 2.2: the strong LTH proves that good subnetworks can **exist** at initialization, but does not address training at all. Existence does not imply trainability: no theorem states that a winning subnetwork embedded in a larger network guarantee good performance. The rest of the network influences its gradient (Section 3).
>
> ### Point 1
> We considered splitting into claim: the lottery analogy is misleading, and alternative view: escape dimensions are a more accurate mental picture. But since the latter is also opinionated, it must be part of our position. We cite in the alternative view section competing, credible, positions. Moreover, not offering a positive intuition would not be constructive.
> *We will make sure to clarify this early on.*
>
> We agree our alternative view section was insufficient and *extensively expanded it with arguments that do not entirely dismiss the LTH, explaining in detail alternative mental pictures and their limitations. For more details, we refer to our response to reviewer `qShV`*
>
> ### Points 2/3
> You are right, this interpretation may not be a leading viewpoint of the community (it is non-trivial to measure this)
> *We will soften 'widespread' in favor of 'a part of the community.'*
>
> Nonetheless, the risk of misinterpretation is **rooted in the name itself**. Using a lottery as analogy conveys a **specific mechanism**: a ticket's success is independent of other tickets, buying more increases your chances, but one winning ticket is enough (buying additional tickets does not affect the outcome of the winning one).
> These are properties of a lottery, not our invention. Their translation to subnetworks is explicitly suggested by Frankle & Carbin's own conjecture. We report three sentences from their paper:
>
> - > "A key question, then, is whether the presence of a winning ticket is necessary or sufficient for SGD to optimize a neural network to a particular test accuracy."
>
>   This leads us to test the **sufficiency** hypothesis.
>
> - > "We conjecture (but do not empirically show) that SGD seeks out and trains a well-initialized subnetwork."
>
>   Proposing the **explanation** that we question in our piece.
>
> - > "By this logic, overparameterized networks are easier to train because they have more combinations of subnetworks that are potential winning tickets."
>
>   This is the core  **scaling** statement that some have picked up and use to explain the success of overparameterization (e.g. the wikipedia quote).
>
> These are not arbitrary: they follow from taking the analogy for what it literally suggests.
>
> ### Point 4
>
> The reviewer finds our evidence for EDT insufficient compared to the LTH's empirical results. As clarified above, we are not competing with those results, but with the LTC that introduced the lottery mechanism, and that some use as an explanation for the success of overparameterization.
>
> EDT is grounded in a body of theoretical work (Fukumizu & Amari 2000, Simsek et al. 2021, ...). The illustrations and simulations are didactical: they make theoretical results accessible and consolidate the EDT mental picture. The proofs are in the cited papers.
>
> ### Conclusion
> The statement of the Frankle & Carbin paper quoted by the reviewer could be called "existence-of-trainable-sparse-subnetworks hypothesis" instead of "lottery ticket hypothesis". However, the impact of the Frankle & Carbin paper goes beyond the empirical findings related to pruning, thanks to the appealing metaphor and the LTC. The LTC bears the burden of comparing large networks with lotteries.
>
> Given our responses, and positive ratings on significance and discussion potential, we ask the reviewer to reconsider their assessment.

---

### Decision · Program_Chairs · 2026-04-30

**Decision:**

Reject

**Comment:**

Based on the reviews, rebuttal, and discussion, I'm proposing a rejection for the paper.

While the reviewers agreed the position of the paper is interesting and could be worth discussing in the community, the discussion surfaced remaining concerns that we believe require a careful revision and an additional round of reviews. I would like to acknowledge that there was not a perfect consensus about the importance to attribute to the concerns below, so the final decision is an unavoidable synthesis of different opinions.

* As it is written, the paper may give the impression of misrepresenting the LTH argument to support the claim that it is "misleading" (as in the title). This is an concern raised by all reviewers and it is an important one. The authors better articulated and nuanced their position in the rebuttal. While some reviewers were positive that simply reworking the "style" of title and some claims in the paper would be sufficient to resolve this problem, other reviewers were concerned that if not properly reflected in a revised paper, the position expressed by the authors could have a negative impact on the community by oversimplifying the discussion for the purpose of supporting their position. This is a very delicate point and based on the discussion I decided that it is more reasonable to wait for a properly revised paper before accpetance.

* LTH and escape dimension seem to study two different aspects of the problem (structural the former, dynamical the latter). This is pointed out both by Rev. dsSd and n2bA. The aspect tends to weaken the logic flow of the paper. In fact, LTH is presented as a flawed tool to interpret overparameterized networks and ED is presented as a better alternative. But if the two approaches fundamentally study different aspects, than ED wouldn't naturally flow anymore as a solution to LTH limitations. I found the authors' rebuttal convincing, but also in this case, it will be crucial for this to be reflected in the paper.

* Empirical evidence is limited. While reviewers in general appreciate the authors to provide empirical support to their claims, the experimental setting does not seem to be representative of the "overparameterized" regime, which severely reduces its effectiveness. Given that this is a position paper and not a full technical contribution, I tend to be lenient on this, but I would still encourage the authors to articulate the value and limitations of the current experiments and also use it as an opportunity to call the community to build on this particular evidence towards stronger empirical investigation.